# The Impact of the COVID-19 Pandemic on Mental Health Care of Children and Adolescents in Switzerland: Results of a Survey among Mental Health Care Professionals after One Year of COVID-19

**DOI:** 10.3390/ijerph19063252

**Published:** 2022-03-10

**Authors:** Anna Maria Werling, Susanne Walitza, Stephan Eliez, Renate Drechsler

**Affiliations:** 1Department of Child and Adolescent Psychiatry and Psychotherapy, University Hospital of Psychiatry Zurich, University of Zurich, 8032 Zurich, Switzerland; susanne.walitza@pukzh.ch (S.W.); renate.drechsler@kjpd.uzh.ch (R.D.); 2Swiss Society for Child and Adolescent Psychiatry and Psychotherapy, 3008 Bern, Switzerland; stephan.eliez@unige.ch; 3Neuroscience Center Zurich, University of Zurich and ETH Zurich, 8057 Zurich, Switzerland; 4Department of Psychiatry, Faculty of Medicine, University of Geneva, 1205 Geneva, Switzerland

**Keywords:** mental health care, treatment supply, children, adolescents, COVID-19, child and adolescent psychiatry, telemental health

## Abstract

Background: To assess the impact of the COVID-19 pandemic on treatment demand and supply in children and adolescents with mental disorders during the first year of the pandemic from the perspective of child and adolescent psychiatrists and psychologists in Switzerland. Methods: The survey was conducted anonymously, in German or French and online in April/May 2021. Mental health professionals working in child and adolescent psychiatry, psychotherapy services or independent practices were contacted by email. Results: N = 454 professionals completed the survey (176 child and adolescent psychiatrists and 276 psychologists). After an initial period of decreased demand during the lockdown in spring 2020, requests for treatment increased, considerably exceeding the demand pre-pandemic and reaching a peak in January/February/March 2021. The vast majority of professionals (78.2%) estimated that there was currently too little supply during the pandemic, which differed from the evaluation of the pre-pandemic situation (37%). A total of 65% of participants indicated that waiting time until the initiation of treatment increased during the pandemic, 41% reported their current workload to be somewhat higher and 44.5% much higher. Conclusions: For the first pandemic year, youth mental health professionals reported a large increase in the treatment demand and waiting time and a worrisome overload of treatment services.

## 1. Introduction

The COVID-19 pandemic has had a tremendous impact on the everyday life and mental health of children and adolescents. The pandemic has caused stress, anxiety, depression, and increased risky or social behavioral problems [1,2], and the rates of suicidal ideation and suicide attempts in young people have risen [3,4,5]. Nevertheless, the findings are not always consistent: studies conducted shortly after the first lockdown (spring/summer 2020) often report a decrease in psychopathological symptoms or in treatment demand (e.g., [6]), but with the publication of more recent studies, the need to take into account the dynamics of the pandemic and the precise timing of research is becoming evident. According to a recent meta-analysis, the prevalence of clinically elevated anxiety and depressive symptoms in children and adolescents has increased in course of the pandemic, with a higher symptom prevalence in studies conducted at the end of the first year than at the beginning [7]. Ordinarily, rising prevalence rates of mental health problems in children and young people should be reflected in higher admission rates to mental health services. Studies are only just beginning to investigate whether this was also the case under pandemic conditions, where possible treatment barriers include social distancing rules, hesitancy to seek help due to fear of infection, and increased institutional problems. A recent European survey among the heads of university departments for child and adolescent psychiatry, encompassing 64 contributions from 22 European countries, including Switzerland, reported a decrease in referrals in April/May 2020 (60% of responses) and a marked increase in February/March 2021 (90.5% of responses). For spring 2020, only a small proportion of respondents (generally less than 30%) reported an increased prevalence occurrence of certain disorders, such as anxiety or obsessive compulsive disorders. By contrast, for February/March 2021, the vast majority of respondents reported increased numbers of cases with anxiety disorders (70%), depression (>60%), eating disorders (>60%), and suicidal crises (>80%) [8]. Data from the Republic of Ireland [9] indicate that referrals to specialized child and adolescent mental health services dropped by around 10% up to August 2020 and then steadily increased up to December 2020, reaching 180% compared to previous years, with double the number of outpatient appointments compared to previous years. A recent qualitative U.S. study reported that services suspected an increased need for mental health care for children and young people during the pandemic, but that staff shortages and decreased service capacity under COVID-19 conditions made it difficult to estimate the true amount [10]. Another U.S. study analyzed admission to a child and adolescent psychiatry inpatient unit between March 2020 and January 2021 and found that 53% of admissions due to adolescent psychiatric crises were related to COVID-19 stressors, although the overall number of admissions had slightly decreased [11].

As another effect of the pandemic, the implementation of telemental health increased, although a number of barriers were identified, such as unclear funding for services, limited internet access, family/client reluctance, or problems linked to online sessions with younger children [10]. A Canadian survey of mental health professionals working in services for mentally ill children and adolescents found that 71% of respondents began to use virtual care during pandemic. The majority of respondents (61%) indicated that virtual care was more effective with adolescents than with children. Most professionals found that telemental health was more tiring and demanded more concentration than in-person sessions and reported that certain relevant non-verbal aspects of the interaction were lacking [12]. Other possible disadvantages of telemental health were the uncertain privacy and the higher attentional demands, which were often too difficult for younger children [13].

The present study aims to assess the impact of the COVID-19 pandemic on children’s and adolescents’ mental health, on the treatment for children and adolescents with mental disorders, and on care service providers during the first year of pandemic from the perspective of youth mental health professionals. In the present analysis, we focus on treatment supply, treatment conditions, and the mental health professionals’ situation during COVID-19. More specifically, we hypothesize that the professionals would report an increased demand for treatment in the long term and extended waiting times until the initiation of treatment in children and adolescents. In addition, we seek to assess the burden and challenges experienced by mental health professionals working with children and adolescents in Switzerland during the pandemic. Since the pandemic made the delivery of in-person psychotherapy difficult or impossible, we expect a sizable number of mental health professionals to recur to telemental health, which, before the pandemic, was very rarely used in child and adolescent psychiatry and psychotherapy in Switzerland. 

## 2. Materials and Methods

The survey was developed as a follow-up and complement to surveys for patients and their parents on the impact of the pandemic on mental health and media use [14,15]. After a literature search, topics and a first set of items were presented to and discussed in a group of mental health professionals working in clinical or outpatient settings. Feedback and suggestions were collected and integrated. A second revised version of the survey was tested and commented by five experienced mental health professionals, which led to the final version. The survey was conducted as an anonymous online survey from 22 April to 24 May 2021. Mental health professionals specialized in the treatment of children and adolescents—i.e., child and adolescent psychiatrists or psychologists—were contacted by email. Addresses were generated from authorized mailing lists of professional societies or via publicly available directories of child and adolescent psychologists and psychiatrists in Switzerland. In total, N = 1870 mental health care professionals with a valid email or post address were contacted. The email cover letter described the purposes and aims of the survey and contained the link to the online survey, as well as a link and reference to the Zurich Department of Child and Adolescent Psychiatry and Psychotherapy (CAPP) website, which provided further information and verified the information provided in the email. A reminder email was sent out about two weeks after the first invitation. In addition, the invitation was published on the internal website of the Zurich CAPP. A small group of mental health professionals, for whom no email address was available, were contacted by letter, which contained the link to the survey and the corresponding QR code. The survey was conducted in German or in French according to the language regions in Switzerland. 

The participants responded by selecting one answer from several options. If none of the options seemed appropriate, they could skip the respective item or specify an answer as a free-text comment. These free-text comments were subsequently analyzed and classified according to the principles of content analysis. 

### Pandemic Measures in Switzerland

While many children and adolescents across the globe had to stay at home for several months, receiving online schooling or no schooling at all, restrictions in Switzerland were less severe. With exception of a lockdown period from mid-March to mid-May 2020, primary schools were allowed to offer on-site schooling if COVID-19 protection measures could be implemented. Many schools already resumed in May 2020 with reduced numbers of students and reduced hours of presence. Nevertheless, secondary schools generally continued to rely on digital classes and homeschooling, and most leisure or group activities were not permitted. There was no further complete lockdown in Switzerland after spring 2020, despite a serious second wave with high incidences of infections in autumn 2020.

## 3. Results

### 3.1. Participants

A total of 454 professionals completed the survey. This corresponds to a response rate of 24.3%. The majority were from the German-speaking part of Switzerland, 14.4% were from the French-speaking part, and 0.4% were from the Italian-speaking part (Table 1). Compared to the language distribution in Switzerland (62.1% of the population with German, 22.8% with French, and 8% with Italian as main language), the French speaking part was slightly underrepresented (Italian-speaking participants had responded to the French language survey) [16]. A total of 38.5% of the respondents were child and adolescent psychiatrists, and 54.4% were psychologists specialized in the treatment of children and adolescents (Table 1). The majority of respondents (N = 247) indicated working in independent practices and N = 135 were employed in clinics for child and adolescent psychiatry, among them N = 107 in outpatient departments (Table 1). Therefore, at least 78% of the respondents provided outpatient treatment, either in an independent practice or in a clinic. The survey was administered to child and adolescent psychiatrists and psychologists only. It is possible that pediatricians with additional qualifications in psychotherapy also participated when they were listed as child and adolescent psychotherapists, or that in very few cases the survey was forwarded to collaborators or team members as they have therapeutic training as well.

### 3.2. Treatment Demand and Supply during the Pandemic

When the different time periods of the pandemic were compared in terms of treatment demand, a relevant decrease (indicated by 45.6% of participants) was limited to the time of the lockdown, in spring 2020. All other time periods were marked by an increased demand for treatment. The winter/spring 2021 period clearly emerges as the time with the greatest increase, with 60.1% of participants indicating a much higher demand than before the pandemic (Table 2).

According to 33% of participants, the demand increased by up to 50% more than usual, and 24.9% of participants stated a 50 to 100% higher demand than usual (Appendix A). Moreover, 65.8% of participants reported that patients had to wait longer for initiation of treatment during the pandemic than before (Table 3a). A further 11% of participants indicated that while the waiting time had not changed, this was because they had made adjustments in response to the increased demand (Table 3a). When asked to indicate the mean waiting time in months until initiation of treatment, 48.6% of participants indicated a pre-pandemic waiting time of less than two months, compared to about 20% in March/April 2021. Currently, almost half of the participants (48.5%) reported a waiting time of three months or more, compared to 15.3% before the pandemic (Appendix A). When separately analyzing the responses from mental health professionals working in clinics for child and adolescent psychiatry and those in independent practices, it became evident that, even before the pandemic, clinic waiting times tended to be longer than waiting times in independent practices (22.4% vs. 11.7% more than three months). In March/April 2021, for admission to child and adolescent psychiatry clinics, 29.9% of responses indicated a waiting time of over three months and 26.9% of responses indicated a waiting time of over six months. By contrast, for admission to independent practices, 32.0% of the responses indicated a waiting time of over three months and 14.6% of responses indicated a waiting time of over six months (Figure 1).

We additionally calculated the mean waiting time, including responses from participants who provided both pre-pandemic and March/April 2021 waiting times (N *=* 360). For responses given as a time range (e.g., 2 to 3 months), which was the case in about 40%, the mean value of the range was used. This analysis resulted in a mean waiting time of 1.68 months (SD 1.49) before the pandemic and a mean waiting time of 3.57 months (SD 3.72) in March/April 2021. This difference was highly significant (Wilcoxon Z *=* −14.448, *p* < 0.001). However, it should be noted that the averages do not adequately reflect the wide range of responses. 

When asked whether treatments could not be conducted as originally planned because of the pandemic, 67% responded that this had been the case, and indicated the following reasons (Table 3b): patients cancelled treatments or did not show up as agreed (33%), mental health professionals had to prioritize their cases according to psychological or psychiatric urgency (28%), or therapies had to be cancelled due to COVID-19 restrictions (33%). Others responded that they used or tried to use telemental health instead of on-site sessions (N *=* 23), or that appointments were cancelled because of quarantine measures (N *=* 11) (free-text comments, Appendix A).

Before the pandemic, 37.7% of participants believed that the supply of mental health services for children and adolescents was too low. Under pandemic conditions in March/April 2021, by contrast, over double this number (78.2%) believed the treatment supply to be insufficient (Table 3c). Responses were very similar when analyzing psychiatrists and psychologists separately: 73% of psychiatrists and 82% of psychologists estimated the treatment supply to be too low in spring 2021 (Figure 2). Likewise, when increased treatment demand was analyzed for the two professions separately, 60.4% of psychiatrists and 60.2% of psychologists reported an alarming increase in January/February/March 2021. As a consequence, the indication of highly increased treatment demand and insufficient treatment supply for children and adolescents during the pandemic appears to be independent of professional affiliation or group interests.

### 3.3. The Situation of Mental Health Professionals during the Pandemic

Compared to before the pandemic, 41% of participants reported their current workload to be somewhat higher and 44.5% much higher (Table 4a). For most participants (29.1%), the period from January to March 2021 was associated with the highest workload during the pandemic months (Table 4b).

Participants reported that particularly stressful aspects of their work were the difficulty of communicating with patients while wearing a mask or via telemental health (very or extremely burdensome for 27.3%), the worry of no longer being able to meet the needs of the patients (26.2%), and the accumulation of patients with acute crises (25.3%) (Table 5). In the free-text comments, 40 participants reported that an increased number of cases and having to turn away families in need with no treatment alternatives were major burdens for them (Appendix A). The lack of prioritization of youth mental health and difficulties in understanding the rationale behind certain COVID-19 measures were also mentioned by some participants (N = 11) as stressful.

However, when asked whether there were also any positive aspects of the pandemic, 61% of the participants agreed. With regard to work, the most frequently mentioned positive aspects were the familiarization with digital techniques and methods of telemental health (N = 128). Telemental health and digital technology included the advantages of being able to conduct sessions with all family members more easily than would be the case in on-site meetings, being able to work from home (N = 35), and having the opportunity for continuous education online (N = 28) (Appendix A).

### 3.4. Experience with Telemental Health

Almost 70% of the participants (N = 317) reported currently using telemental health for sessions with patients or their parents (Table 6a). (Among the many terms used for online therapy in the literature, we opted here for telemental health, referring both to psychiatric and psychological treatment; see, e.g., [17]). For the majority of participants (46.9%), telemental health accounted for only a small proportion of their activities, i.e., less than 10% of all treatment sessions were currently conducted online (Table 6a). The largest proportion of participants (41%) could imagine continuing to use telemental health occasionally after the pandemic. However, less than 6% of participants intended to use telemental health regularly in everyday clinical practice, and almost 9% of the participants preferred to refrain from telemental health completely after the pandemic (Table 6b). Problems in funding treatment (quite or absolutely true: 46.9%), the impersonal nature of telemental health (64.8%), and the lack of security/protection of the communication situation (47.8%) were most frequently cited as arguments against the use of telemental health (Table 7). Technical problems as a barrier to telemental health were more often linked to institutional IT issues (36.1%) than to one’s own insufficient skills (22.1%). Among the participants working in outpatient services in departments of child and adolescent psychiatry (N = 106), 35.8% reported some barriers and 32.2% reported major barriers in the IT domain of the institution. Nevertheless, even if all technical, financial, or practical problems were resolved, less than 5% of participants would want to use telemental health for more than 25% of their treatments (Table 6c).

## 4. Discussion

For the first year of the pandemic, youth mental health professionals in Switzerland reported an initial period of decreased demand for treatment during the COVID-19 lockdown in spring 2020, followed by a successive, slight increase in summer 2020, a marked increase in autumn 2020 (“2nd wave”) and a peak in demand in the first quarter of 2021. This peak came with a considerably increased waiting time for the initiation of treatment and a general shortfall in treatment supply for children and adolescents with mental health problems. Notably, as the two professional groups, psychiatrists and psychologists, did not differ in their evaluation of insufficient treatment supply, these findings cannot be explained by specific professional policies. It is important to point out that the increased need for treatment peaked with a certain delay to the course of the pandemic. In Switzerland, the lockdown was confined to a relatively short period at the beginning of the pandemic and without a strict curfew, and its effects on the well-being of children and adolescents with mental health problems were comparatively mild [18]. For many young people, this period may have represented a protected situation of general deceleration, spent with one’s core family, and with fewer academic or social pressures. As indicated by mental health professionals, some patients did not show up for therapy as planned, possibly for reasons of social distancing or because the symptoms of some patients with pre-existing mental disorders, such as social anxiety, may have temporarily improved during the lockdown (e.g., see [15,19]). In addition, during the summer months of 2020, the incidence rates of COVID-19 in Switzerland were low, and there was a general optimism that the pandemic might soon be over. The observation of a delayed but alarming increase in treatment demand up to January/February/March 2021 was confirmed by the analysis of electronic patient records prior and during the COVID-19 pandemic from the emergency outpatient facility of the Department of Child and Adolescent Psychiatry and Psychotherapy in the Psychiatric University Hospital Zurich [20]. These findings also correspond to reports from some other European countries [8,9], although data on the number of outpatient treatments are not consistent. In a survey of European clinics for child and adolescent psychiatry, respondents more often indicated that the number of outpatients decreased (N = 21) rather than increased (N = 10) in February/March 2021, compared to before the pandemic [8]. This may partly be related to national differences in the organization of treatment supply for children and adolescents with mental health problems.

Regarding their own situation, most respondents indicated that their workload peaked in January/February/March 2021, which is not surprising as it parallels increased demands for treatment. The unusual accumulation of crisis/emergency interventions and the fear of being unable to retain the quality of treatment due to work overload were among the most stressful worries. It was also reported that the triage of whom to accept as an emergency and whom to put on a waiting list was extremely burdensome. In contrast, fear for one’s own health or the health of one’s family members was seldom indicated at this stage of the pandemic, although several respondents mentioned an increasing depletion of their own resources. An increased risk of burnout or psychological distress for mental health professionals under COVID-19 was described [21,22], but resilience seems to outweigh burnout tendencies and fatigue in this population [23].

The percentage of almost 70% of respondents who administered online therapies at some time during the pandemic was remarkable, as the use of telemental health, with exception of phone consultations, seemed very uncommon in child and adolescent psychiatry and psychotherapy in Switzerland before COVID-19. This percentage of almost 70% equals that reported in other surveys with mental health professionals during the pandemic (e.g., 70% of users in a survey of U.S. psychologists [24]). The proportion of current online therapies (March/April 2021), however, was relatively low. After a successful shift from in-person to digital therapy, which became necessary under COVID-19, most mental health professionals obviously preferred to return to onsite contacts as soon as possible. Most respondents also indicated that they intended to use telemental health only from time to time in the future for clinical routine purposes, even if all technical or other problems were resolved. Compared to online therapy with adults, the use of telemental health in children and adolescents may present several limitations. Treatment is often less verbal with younger children than with adolescents, and often includes physical activities (e.g., acting together or playing), which cannot be realized via telemental health. In the case of adolescents, privacy aspects may be less well protected (e.g., if parents or siblings enter the room during the session). Telemental health with suicidal adolescent patients may constitute a risk [12] and should be replaced with on-site contacts. On the other hand, an advantage of telemental health, as mentioned by some respondents, lies in the possibility to bring all involved persons together more easily for counseling, e.g., mother, father, school teacher, or special education teacher. The fact that over 60% of respondents working in institutions identified technical problems as barriers to the use of telemental health leaves room for improvement on the institutional IT level. Nevertheless, the large majority of respondents would only prefer to resort to telemental health occasionally in their daily routines.

Taken together, mental health professionals reported an alarming increase in treatment demand, a large increase in the waiting time before the initiation of treatments, and an increasing work overload, all peaking after one year of the pandemic, in January/February/March 2021. Despite the fact that Switzerland, in comparison to other European countries, only had a short phase of lockdown and school closures, children and adolescents had to face an increase in mental health problems. Health care professionals warned that the lack of school routine and other support provided in the school context might particularly affect children and adolescents with pre-existing mental illness or vulnerabilities [25], and it has been shown that, more generally, school closure and home confinement led to psychological consequences, such as fear, anxiety, restlessness, irritability, and others [26]. However, keeping the schools open is obviously not sufficient to guarantee good mental health.

### 4.1. Implications of the Study and Future Directions

The results of the present study indicate that the shortage of mental health care and services of children and adolescents already existed before the pandemic. This gap has considerably widened since then, due to an alarming increase in the need for treatment during the pandemic. While it was imperative to take measures to protect vulnerable groups during the pandemic, too little attention was paid to other vulnerable groups, namely children and adolescents at risk of mental disorders or with pre-existing mental problems. To counteract this problematic situation and to prevent similar developments in the future, actions have to be taken on various levels.

In the short and medium term, psychiatric and psychotherapeutic care in clinics and practices should be expanded even further to increase treatment capacity. Being forced to perform a triage of patients due to limited resources, as described by many professionals in this survey, is challenging and stressful. It is professionally and ethically difficult to decide which one of two patients may be more in need of treatment, especially as seemingly less acute cases can turn soon into emergencies if they are not treated in time. A more systematic networking of mental health professionals including those in independent practice and a system to report available outpatient and inpatient therapy places could be helpful in times of crises, but, of course only, if free capacities are still available. At the same time, services must be adapted more specifically to the needs of children and adolescents, for example, with the expansion of low threshold access for brief crisis interventions, to home treatment services with a focus on the support of families and on the environment of the child or adolescent. It should be avoided that adolescents in acute crisis have to be triaged into adult psychiatry due to a shortage of inpatient treatment places in child and adolescent psychiatry.

In the long term, the focus should be placed on primary and secondary prevention. All professionals working with children and adolescents can make an important contribution here, e.g., teachers, pediatricians and general practitioners or school social workers and counselors. Professional training for the strengthening of mental health and early recognition of mental disorders to prevent psychiatric disorders and chronicity is recommended. Programs to ensure psychological stability or promote resilience, such as sports or recreational programs, should be expanded and made accessible to all—in particular, to socially disadvantaged families.

In the event of a recurrent coronavirus crisis, professionals should be better prepared to offer alternative treatment options to maintain therapeutic/psychiatric care. Access to telemedicine should be offered early and facilitated, both for the patients and for mental health professionals, without having to struggle with technical difficulties or barriers to financing. However, only ensuring funding for online therapies does not solve the problem with younger or suicidal patients not eligible for telemental health.

Children’s and adolescents’ mental health—especially in times of crises—should be treated as a priority. Child and adolescent psychiatrists or psychologist, and not just adult psychiatrists, should be members of advisory bodies to the government. As far as we know, the situation in child and adolescent psychiatry and the impact of the pandemic on children’s and adolescents’ mental health differed considerably from those reported for adult psychiatry in Switzerland.

Finally, better recognition of the work of mental health professionals and better protection in times of pandemic is urgently needed, especially in mental health care for children and adolescents. In normal times, this could mean a remuneration comparable to that of other medical disciplines, during pandemic, e.g., a prioritization of vaccination, just as for other health professions.

Further research should analyze whether or to what extent countries with more liberal COVID-19 policies still have an advantage regarding the mental health outcomes of their youth, compared to countries with strict home confinement measures and prolonged school closures. Keeping the schools open during the pandemic was probably the best alternative for all concerned, but obviously not sufficient to guarantee good mental health for children and adolescents at risk.

### 4.2. Limitations

The study presents several limitations. All responses were explicitly based on subjective impressions of mental health care professionals, not on objective data, and may thus be biased. In addition, information on pre-COVID-19 conditions and the different periods of the pandemic were collected retrospectively and may be subject to cognitive bias. Information on the age and gender of participants, which might influence the responses, was not collected. Additionally, we did not assess the use of pre-pandemic telemental health. The response rate was relatively low, albeit satisfactory, considering the difficult context of an anonymous survey for a group of people with very little time and work overload. The proportion of professions and language regions of the participants seems to roughly be in agreement with the proportion of mental health care professionals in this domain, to the best of our knowledge, but the representativeness of the study remains questionable. Although the survey items were clearly aimed at the changes brought about by the pandemic, other causes for the reported changes cannot completely be ruled out. Finally, we focused here exclusively on the situation of mental health professionals and did not discuss the reasons or the nature of the increased demand for treatment.

## 5. Conclusions

Taken together, after one year of the pandemic, mental health professionals indicated a large increase in the treatment demand for children and adolescents with mental health problems and a dramatic shortfall of treatment supply, which many already deemed to be insufficient before the pandemic. We also conclude that for the comparative analysis of the pandemic effects, it is crucial to consider the timing of the assessment and the timing of measures taken by the respective national/local governments to combat the pandemic. Research on mental health related to a first lockdown should not be confounded with the results on the long-term consequences of the pandemic. The fact that the negative effects of the pandemic on young people’s mental health seem to have accumulated over time should prompt vigilance regarding the possible future long-term effects.

## Figures and Tables

**Figure 1 ijerph-19-03252-f001:**
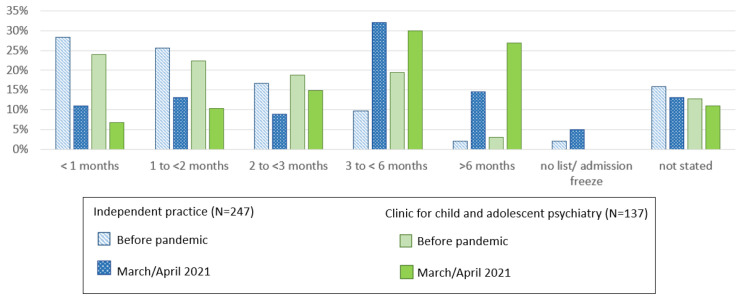
Waiting time until initiation of treatment (months) before the pandemic and currently (March/April 2021) analyzed separately for professionals working in independent practices or in clinics for child and adolescent psychiatry (percentage of responses of each group).

**Figure 2 ijerph-19-03252-f002:**
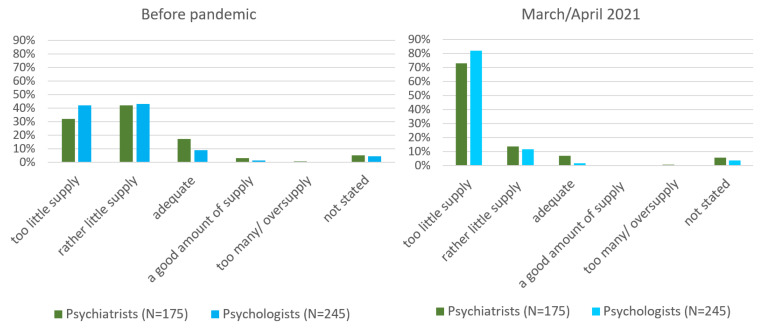
Evaluation of the amount of treatment supply for children and adolescents before the pandemic and in March/April 2021 by psychiatrists and psychologists (percent of responses).

**Table 1 ijerph-19-03252-t001:** Participants.

	N	%
Participant total	454	100%
Professional group		
Child and adolescent psychiatrists	175	38.5
Child and adolescent psychologists	247	54.4
(Other) physician	11	2.4
Nursing/other therapy	7	1.5
Not stated	14	3.1
Work context		
Child and adolescent psychiatric or psychological	247	54.4
independent practice		
Clinic for child and adolescent psychiatry	135	29.8
*outpatient services*	*107*	
*day clinic*	*8*	
*inpatient/ward*	*20*	
Other/not specified	72	15.8
Region		
German-speaking	378	83.2
French-speaking	66	14.4
Italian-speaking	2	0.4
Not stated	8	1.6

**Table 2 ijerph-19-03252-t002:** Number of treatment requests/patient registrations compared to before the pandemic at four different time points (N = 454, percent responses).

	Much Less	Less	Same	More	Much More	Not Stated
March/April/May 2020 (lockdown)	13.4%	32.2%	32.4%	11.5%	3.1%	7.5%
June/July/August/September 2020	1.5%	7.3%	44.3%	34.6%	5.3%	7.0%
October/November/December 2020 ^1^	0.7%	2.4%	15.4%	44.1%	33.5%	4.0%
January/February/March 2021	0.4%	0.9%	11.2%	23.6%	60.1%	3.7%

^1^ = “Second wave”.

**Table 3 ijerph-19-03252-t003:** Treatment waiting time, adjustments, and evaluation of supply in March/April 2021 compared to before the pandemic.

**(a) Current Waiting Time (March/April 2021) for Admission to Treatment Compared to before the Pandemic**				
			**N**	**%**
Shorter than before			8	1.8
Unchanged			60	13.2
Unchanged, but only due to adjustments (e.g., adapted admission criteria, and additional work hours)			54	11.9
Somewhat longer than before			278	61.2
Much longer than before			21	4.6
Not stated			33	7.3
**(b) Modifications of Treatment Intensity or Mode Because of COVID-19 during the First Year of the Pandemic**				
			**N**	**%**
No changes			119	26.2
COVID-related changes occurred (multiple responses possible)			291	64.1
*Cancellation of appointments by patients*			*152*	*33.5*
*Own prioritization according to urgency*			*128*	*28.2*
*COVID-19-related cancellation of treatments (e.g., no group therapy)*			*145*	*31.9*
*Other*			*79*	*17.4*
Not stated/not applicable			46	9.7
**(c) Evaluation of Treatment Supply for Children and Adolescents before the Pandemic and in March/April 2021**				
	**Before Pandemic**		**March/April 2021**	
	**N**	**%**	**N**	**%**
Too little supply	171	37.7	355	78.2
Rather little supply	199	43.8	57	12.6
Adequate, just right	51	11.2	17	3.7
A good amount of supply	10	2.2	1	0.2
A large amount of supply, oversupply	1	0.2	2	0.4
Not stated	22	4.8	22	4.8

**Table 4 ijerph-19-03252-t004:** Workload changes and workload peak during the pandemic.

**(a) Current Personal Workload of Health Professional Compared to before the Pandemic**		
	**N**	**%**
Much lower	2	0.4
Somewhat lower	7	1.5
About the same	40	8.8
A littler higher	186	41.0
Much higher	202	44.5
Not applicable/not stated	17	3.8
**(b) Subjective Peak of Work Demands/Workload across the Last 18 Months**		
	**N**	**%**
January/February 2020 (before lockdown)	6	1.3
March/April 2020 (lockdown)	47	10.4
October/November/December 2020 (2nd wave)	93	20.5
January/February/March 2021	132	29.1
Currently (April 2021)	63	13.9
Not applicable/not stated	113	24.9

**Table 5 ijerph-19-03252-t005:** What was or is particularly stressful/burdensome in your work under pandemic conditions?

	Not Burdensome	Somewhat Burdensome	Fairly Burdensome	Very Burdensome	Extremely Burdensome	Not Stated, Not Applicable
Accumulation of severely affected patients, crisis interventions	8.1%	22.9%	31.3%	21.1%	4.2%	12.3%
Increased stress in everyday life, increased concentration due to COVID-19 rules	9.3%	35.2%	28.9%	15.4%	2.6%	8.6%
Difficulty communicating with mask or via video	5.1%	29.3%	33.7%	18.7%	8.6%	4.6%
Uncertainty of the situation, planning uncertainty	15.4%	36.3%	25.6%	11.2%	3.7%	7.7%
Concern that I might no longer be able to meet the needs of the patients adequately	11.5%	27.5%	25.6%	17.8%	8.4%	9.3%
Worry about infecting my family	35.7%	33.9%	12.3%	8.1%	2.2%	7.7%
Worry about getting infected	37.2%	38.1%	9.0%	5.7%	1.8%	8.1%
Financial loss	57.9%	20.7%	6.2%	2.4%	1.5%	11.2%

**Table 6 ijerph-19-03252-t006:** Use of telemental health.

**(a) Do You Currently Offer Telemedicine/Consultation?**		
	**N**	**%**
Yes	317	69.8
*If yes: the percentage of consultations currently conducted online*		
*Less than 10%*	*213*	*46.9*
*Up to 25%*	*80*	*17.6*
*Up to 50%*	*10*	*2.2*
*More than 50%*	*7*	*1.5*
*Not applicable/don’t know*	*7*	*1.5*
No	84	18.5
Not useful in own work setting (e.g., ward or day clinic)	15	3.3
Not stated	38	8.4
**(b) Where Do You See Benefits/How Do You Intend to Use Telemental Health in the Long Run?**		
	**N**	**%**
I intend not to use it after the pandemic	40	8.8
I intend to use it as an exception, not in clinical routine	151	33.3
I intend to use it from time to time in daily clinical practice	186	41.0
I intend to use it often and regularly in clinical routine	27	5.9
I would like to use telemedicine more often, but see difficulties	8	1.8
Not stated	42	9.3
**(c) If There Were No Technical/Practical Barriers, What Percentage of Your Consultations Would You Like to Conduct Online?**		
	**N**	**%**
None	80	17.6
Less than 10%	210	46.3
Up to 25%	112	24.7
Up to 50%	18	4.0
More than 50%	1	0.2
Not stated	33	7.3

**Table 7 ijerph-19-03252-t007:** Where do you see difficulties in the use of telemental health? (Percent of responses).

	Not True	Hardly True	Do not Know; Neutral	Quite True	Absolutely True	Not Stated
Rejection by patients/parents	19.2%	28.9%	13.4%	24.9%	1.5%	12.1%
Too impersonal/no protected therapeutic space	8.4%	14.5%	4.0%	41.0%	23.8%	8.4%
Health insurance coverage/funding	15.9%	7.0%	11.9%	20.0%	26.9%	18.3%
IT problems (institution)	27.5%	13.4%	4.8%	23.3%	12.8%	18.1%
Own IT knowledge/problems	40.1%	20.5%	4.6%	19.2%	2.9%	12.8%
Security concerns/data protection	13.4%	19.6%	9.9%	33.9%	13.9%	9.3%

## Data Availability

Not applicable.

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
