# Peer review of "The Impact of the COVID-19 Pandemic on Mental Health Care of Children and Adolescents in Switzerland: Results of a Survey among Mental Health Care Professionals after One Year of COVID-19"

_ijerph, 2022, doi:10.3390/ijerph19063252_

Round 1
Reviewer 1 Report
The paper is an interesting article describing a survey conducted by interviewing child and adolescent mental health professionals.
The authors evaluated the access of young patients to facilities (outpatient only) and mental health professionals during the pandemic period of COVID.
The topic is very interesting because there are many data indicating an increase of psychological and psychiatric disturbances during the COVID pandemic in recent scientific literature.
In fact, it is not entirely clear whether there is a temporary increase in anxious and depressive symptoms linked to subthreshold disturbance before the pandemic or an increase in the incidence, especially if these disturbances last over the next few years.
The Authors evaluated if the role of mental health services is sufficient or not.
The authors created a questionnaire that they sent to the mental health professionals, where they explored the number of people who were referred and the judgment given by the health professionals about the possibility of responding to treatment.
They also evaluated the use of the telemedicine tool.
The results were compared with the international literature.
At the beginning of the pandemic period, there was a reduction in access and a subsequent increase.
Telemedicine was seen as a helpful resource but not a complete replacement for face-to-face visits.
The abstract is complete.
The introduction is well done.
The materials accurately describe the research tools.
The results are presented clearly, with the use of tables and graphs.
I think that the authors should indicate whether access to the facilities was due to an increase in cases of anxious or depressive disorders or to an exacerbation of previously diagnosed pathologies.
If the data were not available, it could be added to the limitations.
The data well support the conclusions.
Author Response
Response to Reviewer 1 Comments
The paper is an interesting article describing a survey conducted by interviewing child and adolescent mental health professionals.
The authors evaluated the access of young patients to facilities (outpatient only) and mental health professionals during the pandemic period of COVID.
The topic is very interesting because there are many data indicating an increase of psychological and psychiatric disturbances during the COVID pandemic in recent scientific literature.
In fact, it is not entirely clear whether there is a temporary increase in anxious and depressive symptoms linked to subthreshold disturbance before the pandemic or an increase in the incidence, especially if these disturbances last over the next few years.
The Authors evaluated if the role of mental health services is sufficient or not.
The authors created a questionnaire that they sent to the mental health professionals, where they explored the number of people who were referred and the judgment given by the health professionals about the possibility of responding to treatment.
They also evaluated the use of the telemedicine tool.
The results were compared with the international literature.
At the beginning of the pandemic period, there was a reduction in access and a subsequent increase.
Telemedicine was seen as a helpful resource but not a complete replacement for face-to-face visits.
The abstract is complete.
The introduction is well done.
The materials accurately describe the research tools.
The results are presented clearly, with the use of tables and graphs.
Point 1: I think that the authors should indicate whether access to the facilities was due to an increase in cases of anxious or depressive disorders or to an exacerbation of previously diagnosed pathologies.
If the data were not available, it could be added to the limitations.
The data well support the conclusions.
Response 1: Thank you very much for your constructive feedback.
We agree that information on changes in severity of psychopathological problems during the pandemic or on mental problems triggered by the pandemic would be relevant. We added this point to the limitation section, which now reads (page 12, lines 396ff):
Finally, we focus here exclusively on the situation of mental health professionals and do not discuss the reasons or the nature of the increased demand for treatment.

Reviewer 2 Report
The authors conducted a web-based survey of professionals (i.e. psychiatrists and psychologists) offering child and adolescent mental health care in Switzerland. It was found that in the beginning of the pandemic, the mental health service volume decreased and then gradually increased to the peak in early 2021. The surveyed professionals found that at the peak of the demand of child and adolescent mental health services, the supply was inadequate, the workload was heavy and they might not be able to a good enough job. It seemed that most of the surveyed professionals were not enthusiastic about the use of telemental service in the future.
The research is unique for depicting the situation in Switzerland through the perspectives of the surveyed professionals, however, there are following issues that the authors need to handle before their manuscript is publication worthy:
- The authors should have described how they developed their survey questionnaire.
- The representativeness of the sample should have been clarified in more details. For example, what was the population size of the professionals offering child and adolescent mental health services? How many were invited for the survey? What was the response rate?
- On page 3 lines 100-110, it was stated that the invited surveyees were psychiatrists or psychologists offering child and adolescent mental health services. But, in Table 1, some surveyees were other physicians, nurses, other therapy professionals, and those who did not state their professions. It is better the authors explain why this happened and address how to handle the data offered by these surveyees with justified explanations.
- Based on the manuscript, we do not know how the surveyees estimated the change of the demand and waiting time for mental health services (pre-pandemic v. post-pandemic). The authors should have stated whether the surveyees were required to check their practice data or could just offer their subjective impressions? This may limit the reliability and validity of the survey results.
- On page 5, It is better to merge figures 1A and 1B together to show more clearly the contrast before and after the pandemic.
- On page 5, line 189, how could “to prioritize more urgent cases” be related to the pandemic? Does it mean the professionals had to take care of the COVID-19 patients or resources were moved from mental health care to the COVID-19 care?
- The title of Table 3 should have been revised to reflect that the contents were related to the changes before and after the pandemic. On sub-tables 3-1 and 3-2, it is not very clear which two time periods were compared. The authors should have specified the time periods clearly both in the text and in the sub-tables.
- On page 8 line 223, please clarify what “do justice to the patients” means in the manuscript.
- On page 10 line 286, the wording “winter 2021” was confusing. Please specify both the months and the year.
- In the Discussion Section, the authors should have addressed the limitation of the research.
- The authors could try to propose a theory to explain the changed patterns of child and adolescent mental health service and what next to do to for further investigation.
- Although the title of the manuscript carries the wording “Impact of the COVID-19 pandemic on mental health care of 2 children and adolescents in Switzerland”, actually the survey method only addressed the “change” before and after the COVID-19 pandemic. The authors should have specified that the “impact” was inferred quite indirectly or replace “impact” with another appropriate word.
- Based on the manuscript text, it is not clear that the results about telemental health services could reveal what “impact” the COVID-19 pandemic had on the its use.
- It would be better if the authors could address more the policy implications of their findings and how they could be similar to or different from the ones in other countries.
Reviewer 3 Report
Important investigation, I highly recommend publication. However I have a few comments that I hope would be helpful.
You have 454 mental health professionals filling your survey. Out of how many contacted? Furthermore, how much of the mental health landscape does it represent?
You say that most answers were from the German speaking population. Is that because they were more targeted? Is there geographical data that can be shared? Maybe a map of Switzerland where these findings apply.
Are there response biases due to career length of the health professional? Are there response biases due to gender? If this data is available to you, it would allow us to draw an important picture of how mental health changed during the pandemic and how the mental health system perceived it.
Your conclusions are based professional reports of how much supply/demand in treatment has changed. I suppose most of these practices have the actual numbers of registrations/admissions that change over time and that they can share. This would provide a more fine grained analysis of the situation and could help tune new social policies and maybe prevention measures.
Your findings are important. I would enjoy and I believe other readers would also enjoy if you extend your discussion about the implications of your findings, potential alternative solutions that could help with remote and noninvasive detection of mental health status, to unload some of the burdens from the mental health system and improve patient treatment time and quality.
